**Data Availability Statement:** All FASTA input files and analysis datasets used in this paper are available at the Harvard Dataverse repository (https://doi.org/10.7910/DVN/Q6HVFO).

**Funding:** All authors were supported by National Institutes of Health grant 5R01AI100947–06, "Algorithms and Software for the Assembly of

# Measuring quality of DNA sequence data via degradation

**Alan F. Karr**[1]*, **Jason Hauzel**[1]¤, **Adam A. Porter**[1,2]☯, **Marcel Schaefer**[1]☯

1 Fraunhofer USA Center Mid-Atlantic, Riverdale, MD, United States of America, 2 Department of Computer Science, University of Maryland, College Park, MD, United States of America

☯ These authors contributed equally to this work.
¤ Current address: Capital One, Washington, DC, United States of America
* akarr@fraunhofer.org

## Abstract

We formulate and apply a novel paradigm for characterization of genome data quality, which quantifies the effects of intentional degradation of quality. The rationale is that the higher the initial quality, the more fragile the genome and the greater the effects of degradation. We demonstrate that this phenomenon is ubiquitous, and that quantified measures of degradation can be used for multiple purposes, illustrated by outlier detection. We focus on identifying outliers that may be problematic with respect to data quality, but might also be true anomalies or even attempts to subvert the database.

## Introduction

As public genome databases proliferate, their immense scientific power is tempered by skepticism about their quality. The skepticism is not merely anecdotal: there are documented instances and implications [1–3]. Although we argue in S1 Appendix that data quality should not be construed as comprising only errors in data, the principal contribution of the paper is a novel paradigm for measuring quality of genome sequences by deliberately introducing errors that reduce quality, a process we term degradation. The errors are single nucleotide polymorphisms (SNPs), insertions and deletions that both occur naturally as mutations and arise in next generation sequencing. Our reasoning is that *higher quality data are more fragile*: the higher the initial quality, the greater the effect of the same amount of degradation. We present evidence that supports this reasoning, as well as demonstrates the scope and consequences of the phenomenon.

Even though the main contribution of the paper is methodological, applicability to bioinformatics problems is its *raison d'être*. Our exemplar problem is detection of outliers in genome databases. We identify genomes in a 26,953 coronavirus database downloaded from the National Center for Biotechnology Information (NCBI), whose degradation behavior is anomalous, and whose quality, therefore, may be suspect. We detect deliberately inserted low quality genomes, but other genomes in the original database are equally problematic. A second potential application is to thwart adversarial attacks on genome databases that, for instance, insert artificial genomes so that sequences of concern such as those generated by the methods

Metagenomic Data," to the University of Maryland College Park (Mihai Pop, PI), via a subaward to Fraunhofer USA. The sponsor URL is www.nih. gov. The sponsor played no role in the research, decision to publish, or preparation of the manuscript.

**Competing interests:** The authors have declared that no competing interests exist.

in [4] will pass screening tests. Finally, degradation can be used to characterize the quality of synthetic DNA reads that are used to evaluate genome assemblers [5].

## Materials and methods

Our method is rooted in total quality paradigms for official statistics, that is, censuses and surveys conducted by national statistics offices (S1 Appendix). In that context, data quality is a longstanding issue, and low quality data are known to be resistant to further errors, such as those introduced by editing, imputation or statistical disclosure limitation (SDL). We also draw on official statistics for techniques to quantify data quality. In experimental settings and because it is intuitive, we measure degradation by distance, appropriately defined, from the stating point. In real databases, this is not possible, so we employ measures of distance from a universal "endpoint" representing the lowest possible quality—pure randomness in the form of maximal entropy, which every genome reaches in the limit of infinite degradation.

### Preliminaries

In this paper, a genome $G$ is a character string chosen from the alphabet $\mathcal{B} = \{A, C, G, T\}$, and represents one strand of the DNA (or, for viruses, RNA. in an organism. The constituent bases (nucleotides) are A = adenine, C = cytosine, G = guanine and T = thymine. We denote the length of a genome $G$ by $|G|$; the $i^{\text{th}}$ base in $G$ is $G(i)$; and the bases from location $i$ to location $j > i$ are $G(i\colon j)$. Given an integer $n \geq 1$, the $n$-gram distribution is the probability distribution $P_n(\cdot|G)$ on the set all subsequences of length $n$ chosen from $\mathcal{B}$—there are $4^n$ of them—constructed by forming a table of all length $n$ contiguous substrings of $G$ and normalizing it so that its entries sum to 1. There are $|G| - n + 1$ such sequences, starting at $1, 2, \ldots, |G| - n + 1$, so the normalization amounts to division by $|G| - n + 1$.

In this paper, we focus on *triplets*, which are 64-dimensional summaries of genomes, and which also encode amino acids—the building blocks of proteins. The interpretation of $P_3(\cdot|G)$ is that for each choice of $b_1, b_2, b_3$ from $\mathcal{B}$,

$$P_3(b_1 b_2 b_3 | G) = \text{Prob}\{G(k : [k + 2]) = b_1 b_2 b_3\}, \tag{1}$$

where $k$ is chosen at random from $\{1, \ldots, |G| - 2\}$. Triplets provide a generative model of a genome as a second-order Markov process, since $P_3$ contains the same information as the pair distribution $P_2$ and the $16 \times 4$ transition matrix

$$T_3(b_1, b_2, b_3 | G) = \text{Prob}(G(k + 2) = b_3 | G(k) = b_1, G(k + 1) = b_2) \tag{2}$$

that gives the distribution of each base conditional on its two immediate predecessors.

Distributions of bases, pairs of successive bases, triplets of successive bases and quartets of successive bases differ across genomes, in ways that support a variety of analyses, including not only outlier identification, which we address below, but also Bayesian classification of simulated next generation sequencer reads and detection of contamination [6]. How tuple distributions behave under degradation supports our hypothesis regarding data quality. Higher-level genome structure such as repeats and palindromes is addressed in the below.

Other cases of interest are less suited to our purposes. Base ($n = 1$) and pair ($n = 2$) distributions are too coarse to be useful on their own. Quartets ($n = 4$) have been studied extensively [7–9]. For the problems we address, they are more cumbersome than triplets without being significantly more informative. Finally, although we do not do so here, triplet distributions can usefully be converted to amino acid distributions [6].

## The hypothesis and initial evidence

We hypothesize that because high quality data are more fragile than low quality data, As noted, there is precedent in official statistics for this assertion. Some components of the argument appear in S1 Appendix, while the total survey error (TSE) paradigm referred to in S1 Appendix rests in part on this premise.) the quality of elements of a DNA sequence database can be measured by degrading them, Secondarily, the more complex the characteristic examined, the greater the impact of degradation. As we see below, the effect of degradation increases as we move from base distributions to pair distributions to triplet distributions to quartet distributions to repeats and palindromes.

We perform the degradation by iteratively applying the `Mason_variator` software [10]. Briefly, the `Mason_variator` simulates changes to a genome sequence: SNPs, insertions, deletions, inversions, translocations, and duplications, with specified probabilities for each. Such changes occur naturally as mutations as well as in reads produced by next generation sequencers, such as those manufactured by Illumina. `Mason_variator` runs from a command line interface with user-specified parameters, input files, and output files. For simplicity, in most of our analyses, only SNPs were simulated. The principal reason is to avoid burdensome computation of Levenshtein distances. Iterative use of `Mason_variator` means starting with a given genome, running `Mason_variator` on it, running `Mason_variator` again on the result, . . ., up to a specified number of iterations, which is usually 2000. Much the same effect could be achieved by increasing the error probabilities, but at a loss of interpretability, because parametrization by the number of iterations is more intuitive. Evidently this process of iteration is analogous to the real process of evolution.

We have investigated several measures of quality for degraded genomes. The first two of these are employed in string matching: Hamming distance [11] is usable when only SNPs are simulated, while Levenshtein distance [11] allows insertions and deletions that alter the length of the DNA sequence. (The Hamming distance between sequences with different lengths is infinite. Levenshtein distance is significantly more burdensome than Hamming distance computationally, with respect to both time and memory requirements, especially for longer genomes.) As discussed below, the origin point for Hamming and Levenshtein distances is crucial. We treat distances based on distributions of nucleotides, pairs, triplets and quartets as well as entropy of triplet distributions of degraded genomes.

Fig 1 visualizes the hypothesis for a single element of the coronavirus database employed in our primary experiments. All forms of errors were allowed. In the figure, the $x$-axis is the number of iterations of the `Mason_variator`, and the $y$-axis is Levenshtein distance between the degraded genome and the original genome. The most salient characteristic of the curve is its concavity: the more degradation already done, the less the effect of each additional iteration.

There are issues with the choice of the origin for Levenshtein distances. In Fig 2, there are 21 initial genomes—the one randomly selected coronavirus genome and that genome after 100, 200, . . ., 2000 `Mason_variator` iterations, representing continually decreasing initial data quality. In the top panel, Levenshtein distance is measured from the parent (0-iteration) genome, and the distance at iteration 0 has been subtracted from each curve. In a sense, however, this is "cheating," because in real databases that potentially contain errors, there are not definitive parent genomes. In the middle panel in Fig 2, Levenshtein distance for each curve is measured from its starting point. The curves there differ little, and certainly not systematically. Fortunately, a work-around exists: the bottom panel in Fig 2 shows that (within reason), any fixed genome can be used as the origin. There, all Levenshtein distances are measured from a second randomly selected genome in the NCBI dataset. The key point is that the

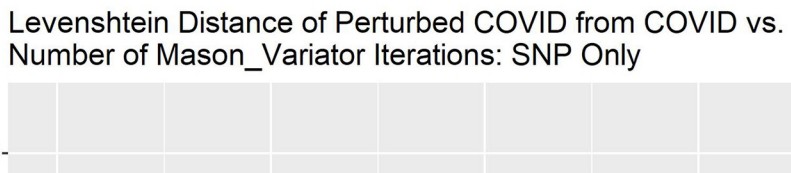

**Fig 1. Examplar of degradation.** Levenshtein distance from a (randomly selected) coronavirus genome as a function of the number of `Mason_variator` iterations.

curves in the bottom panel vary significantly and systematically with respect to the initial degradation.

## Experimental platform

Our experimental platform is a coronavirus database containing 26,953 genomes, which was downloaded from the NCBI in November of 2020. To it, we added eleven "known" problem cases: a single adenovirus genome with length 34,125 BP (downloaded as part of the `Art` read

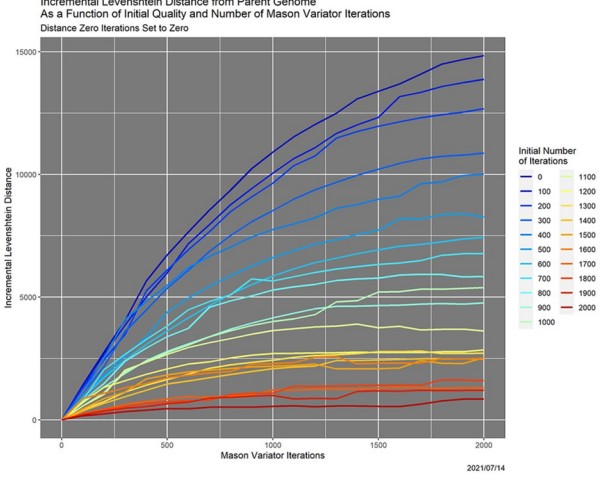

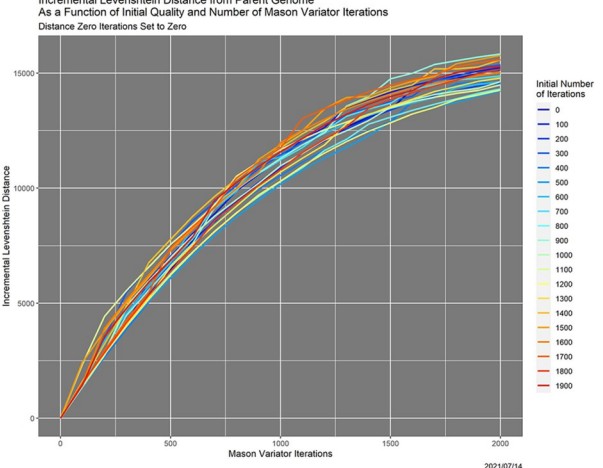

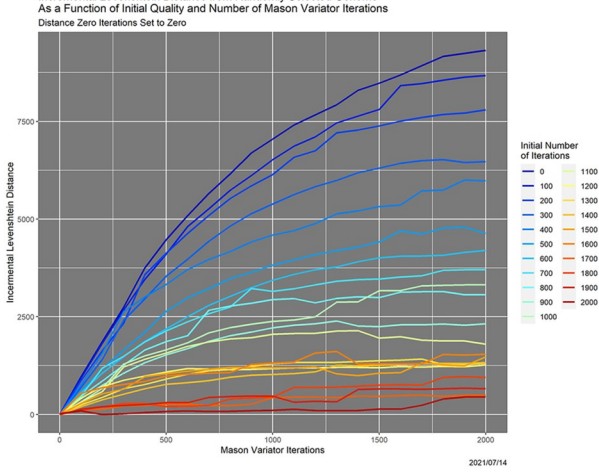

**Fig 2. Degradation behavior as a function of `Mason_variator` iterations and initial data quality, for a randomly selected coronavirus genome.** A: Degradation is measured by incremental Levenshtein distance from the parent (0-iteration) genome. B: Degradation is measured by Levenshtein distance from the starting point. C: Degradation is measured by incremental Levenshtein distance from a second randomly selected genome in the NCBI database. Color encodes the initial number of `Mason_variator` iterations.

simulator software package [12]) and low-quality versions of 10 coronavirus selected randomly from the original 26,953, each created by 2000 iterations of the `Mason_variator`. Our method detects not only these known outliers—the minimal criterion for credibility, but also others.

All FASTA input files and analysis datasets used in this paper are available at https://dataverse.harvard.edu/dataset.xhtml?persistentId=doi:10.7910/DVN/Q6HVFO. The human genome file there contains only the four sequences (chromosomes 1, X, Y; mitochondrial) appearing in Fig 10. The full file is available at https://www.ncbi.nlm.nih.gov/assembly/GCF_000001405.39, but has been superseded by https://www.ncbi.nlm.nih.gov/assembly/GCF_000001405.40.

## Measuring degradation

**General effects.** For the adenovirus genome, Fig 3 shows the effect of degradation on the base, pair, triplet and quartet distributions, measured by Hellinger distance [13] from corresponding distributions for the original genome. The interpretation is that as the number of `Mason_variator` iterations increases, base, pair, triplet distributions, and quartet distributions all move farther and farther away from the parent genome, at slower and slower rates.

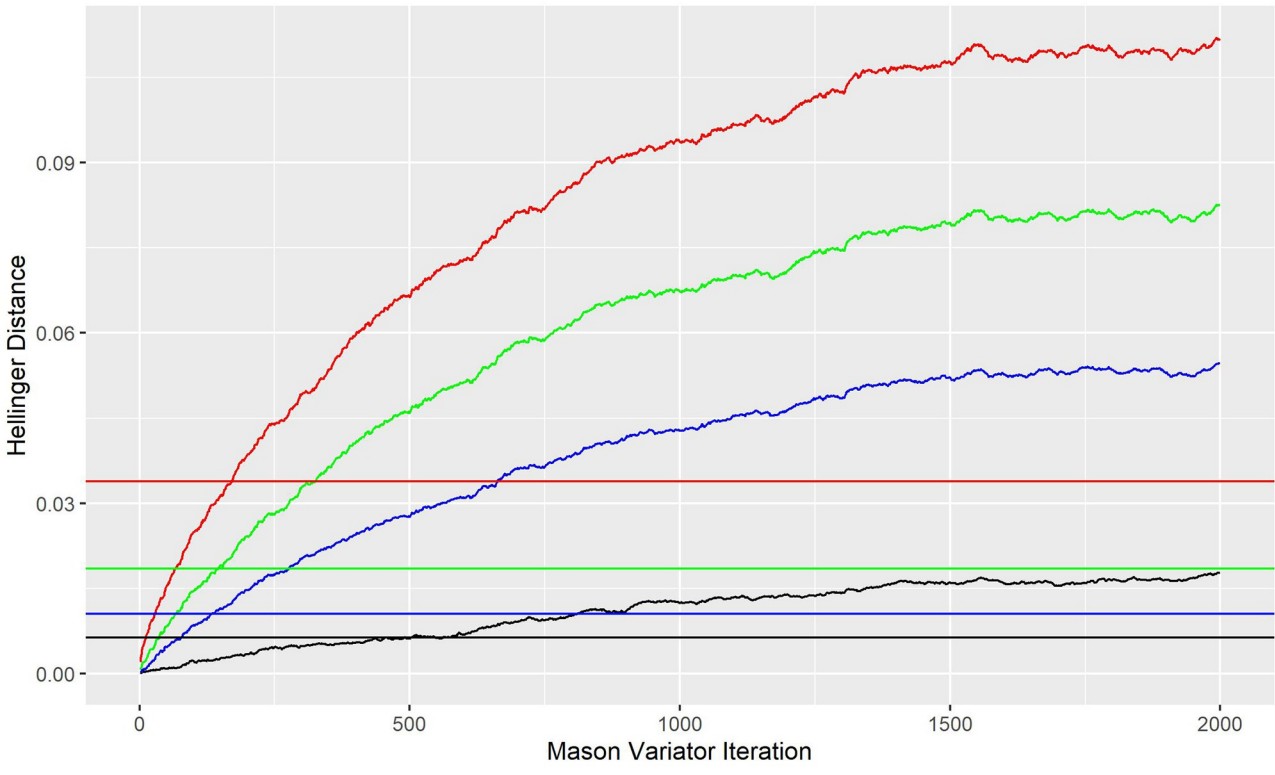

**Fig 3. Effect of degradation on tuple distributions.** Shown are the base distribution (black), pair distribution (blue), triplet distribution (green) and quartet distribution (red) for the adenovirus genome.

Moreover, confirming our secondary hypothesis, the higher the dimensionality, the more rapid the movement: quartets are more fragile than triplets, which are more fragile than pairs, which are more fragile than individual bases. The horizontal lines in Fig 3, whose colors match those of the curves, are simulation-derived 1% p-values: the probability that the distribution matches that of the original genome is less than 0.01 when the distance exceeds the line. Interestingly, the numbers of iterations at which the 1% thresholds are passed (i.e., where the curves cross the lines) are nearly the same for pairs, triplets and quartets, but lower than for bases alone.

**Measuring degradation of triplet distributions via entropy.** So far, we have confined attention to what degradation moves away from. In many ways, what it moves *toward* is more useful, because there is a single infinite degradation endpoint representing maximum entropy. The entropy of a probability distribution $P$ on a finite set $\mathcal{S}$ is

$$H(P) = -\sum_{s \in \mathcal{S}} p(s) \log p(s), \tag{3}$$

with the convention that $0 \times -\infty = 0$. Entropy is minimized by distributions concentrated at a single point and maximized at the uniform distribution on $\mathcal{S}$, with maximizing value $\log(|\mathcal{S}|)$, where $|\cdot|$ denotes cardinality. The existence of the universal maximizing value enables us to measure degradation as *movement toward maximum entropy*, removing the common origin issue discussed noted previously.

Fig 4 shows the effect of 500 `Mason_variator` iterations on entropy of triplet distributions—hereafter, just triplet entropy—of the adenovirus genome, starting from the genome itself (black curve) compared to starting from the genome degraded by 250 `Mason_variator` iterations (blue curve), degraded by 500 `Mason_variator` iterations (green curve), degraded by 1000 `Mason_variator` iterations (yellow curve), and degraded by 1500 `Mason_variator` iterations (red curve). The $y$-axis is the increase in entropy as a function of `Mason_variator` iterations, so Fig 4 shows movement toward maximal entropy.

## Results

### Full dataset

Fig 5 shows, albeit with massive overplotting, the triplet entropy degradation for the entire 26,964-element experimental database. The adenovirus genome, in blue, and the 10 degraded coronavirus genomes, in red, are apparent outliers. But, clearly there are also other outliers, which we pursue momentarily.

### Outlier detection

One effective strategy for identifying elements of a genome database with problematic quality is to search for outliers. We concede that the implicit presumption that the bulk of the database is of high quality may be untested. The key question is, "Outlying with respect to what metric?" In this section, the metric is based on hierarchical cluster analysis of triplet entropy increase resulting from `Mason_variator` degradation, that is, on the shapes of the curves in Fig 5. The clustering is in three dimensions, as opposed to 64 dimensions for triplet distributions and 21 for amino acid distributions. Possibly unexpectedly, the two sets of clusters are very similar.

**Outlier detection using triplet distributions.** We showed in [6] that clustering of triplet distributions identifies outliers. Briefly, we performed hierarchical clustering, using Euclidean distances and "complete clustering" in R [14], on the 26,964-genome database, using as clustering variables the 64 standardized components of the triplet distributions. By means of standard

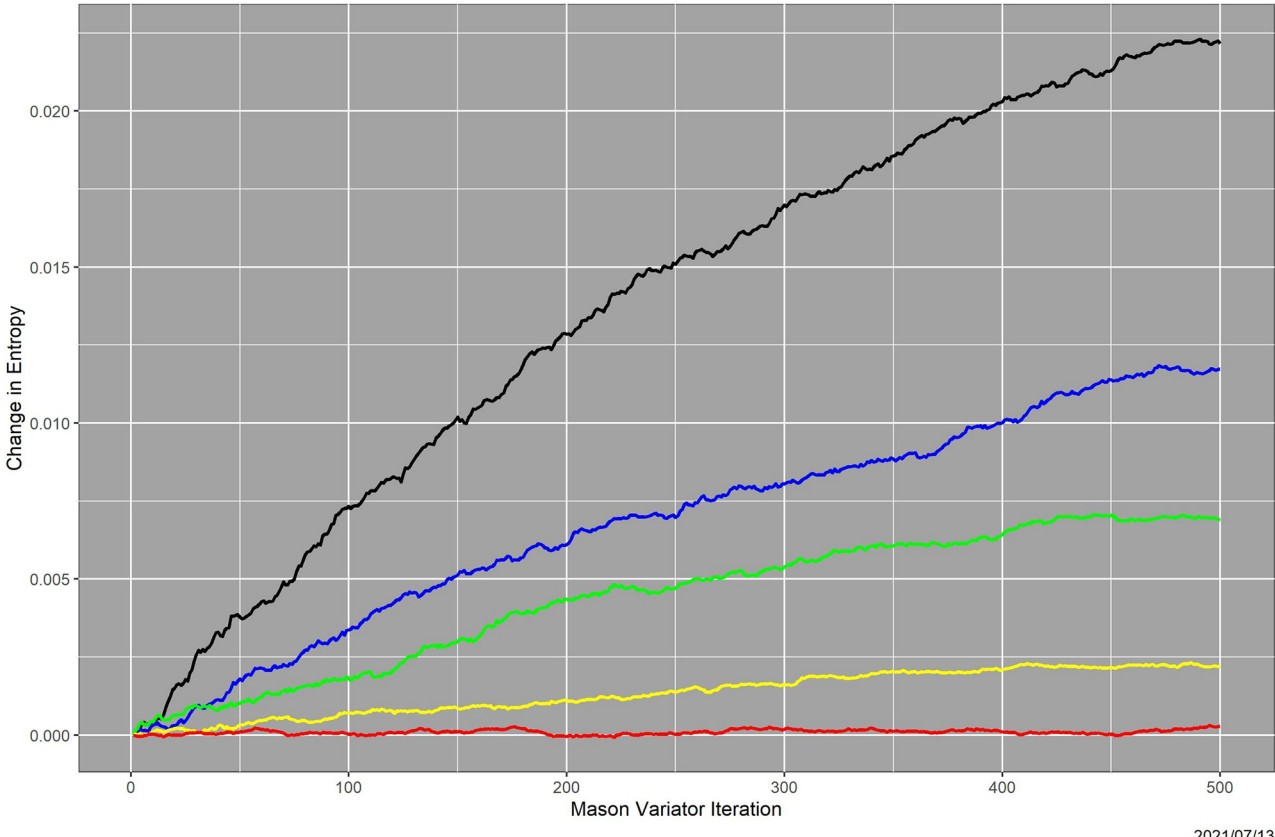

**Fig 4. Change in entropy as a function of `Mason_variator` iterations, for one adenovirus genome and four degraded versions of it.** *Black*: original genome; *Blue*: degraded by 250 iterations; *Green*: degraded by 500 iterations; *Yellow*: degraded by 1000 iterations; *Red*: degraded by 1500 iterations.

heuristics that trade off model fit and model complexity, the number of clusters was determined to be 23.

Fig 6 is a plot of two-dimensional multidimensional scaling (MDS) [15, 16] of the 23 cluster centroids. The overwhelming majority of coronavirus genomes—26,433 of the original 26,953, or 98.1%—are in a single cluster. One original coronavirus genome appears by itself, in cluster 12. Cluster 13 contains the adenovirus genome alone, while each of the 10 degraded coronavirus genomes appears in a cluster by itself (clusters 14–23). Thus, the deliberate outliers are not only detected but also distinguished from one another. The dendrogram in Fig 7 shows that the coronavirus genome in cluster 12 and the deliberate outliers are separated from the remaining 26,952 coronavirus genomes at the first split in the clustering process. Clusters 1–10, which are small, are potential outliers as well. See [6] for details and a scientific interpretation.

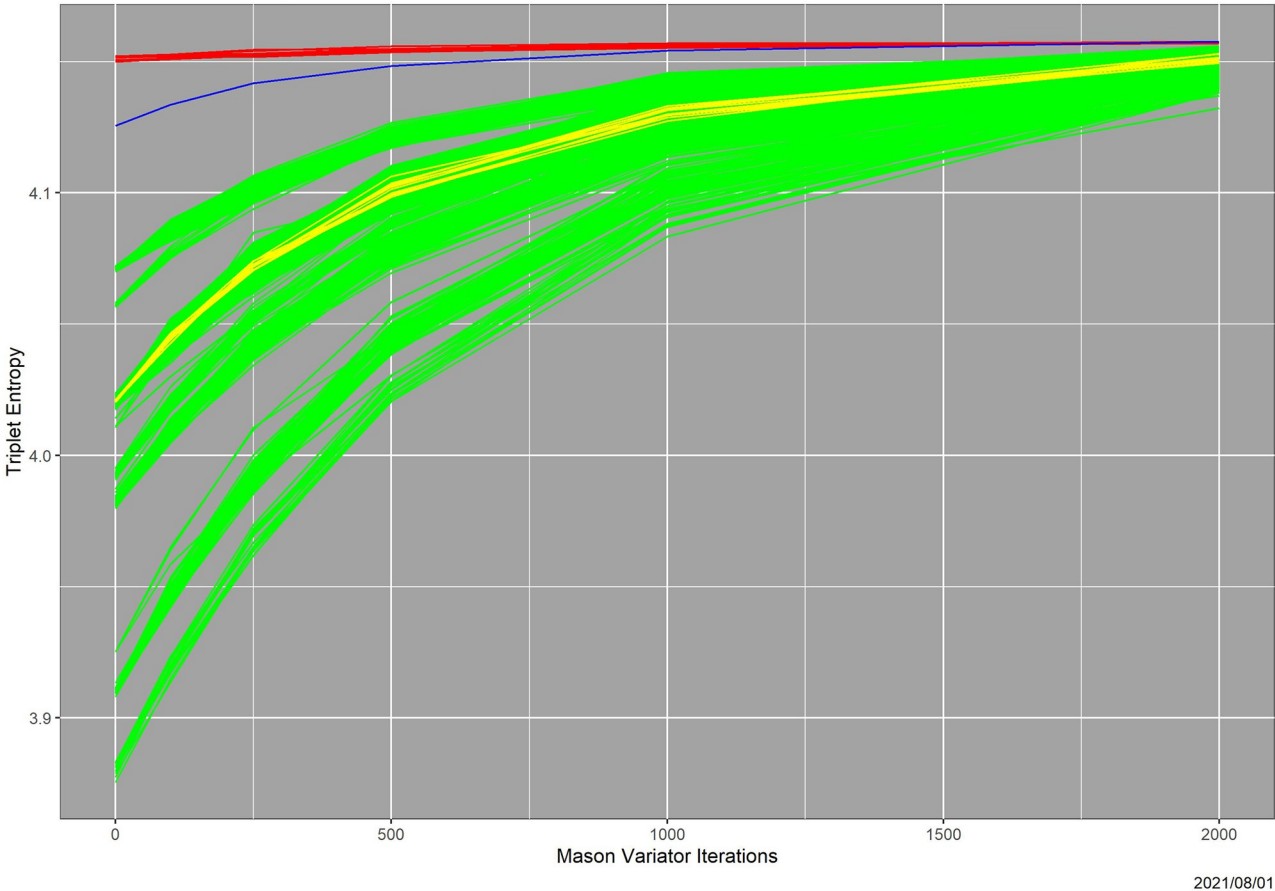

Fig 5. **Entropy as a function of `Mason_variator` iterations, for the 26,964-genome dataset.** The adenovirus outlier is in blue, the 10 degraded coronavirus outliers are in red, and the parents of the ten coronavirus outliers are in yellow.

**Outlier detection using entropy degradation.** Here we cluster the genomes on the basis of degradation behavior of triplet entropy, that is, the curves shown in Fig 5. As noted already, the results match closely with those using triplet distributions.

The clustering is now in only three dimensions, reached by a path that starts with Fig 5. Every one of the 26,964 curves plotted there is based on entropy following 0, 250, 500, 1000 and 2000 `Mason_variator` iterations. We fitted a quadratic function to each set of 5 values, reducing the dimension to 3. These quadratic models are uniformly good: the smallest of the coefficients of determination, $R^2$, is 0.941 and 99% of them exceed 0.976. Hierarchical clustering was then performed on standardized versions of the three quadratic coefficients, using the "ward.D" option in R, resulting in 34 clusters, with counts ranging from 11 to 1470. Statistically, the clustering is extremely good: the cluster numbers alone explain 98.96509%, 99.02426% and 98.45677% of the variation in the quadratic coefficients.

Paralleling Figs 6–9 show the result of applying two-dimensional MDS to the cluster centroids, as well as the associated dendrogram. There is nothing comparable to the massive

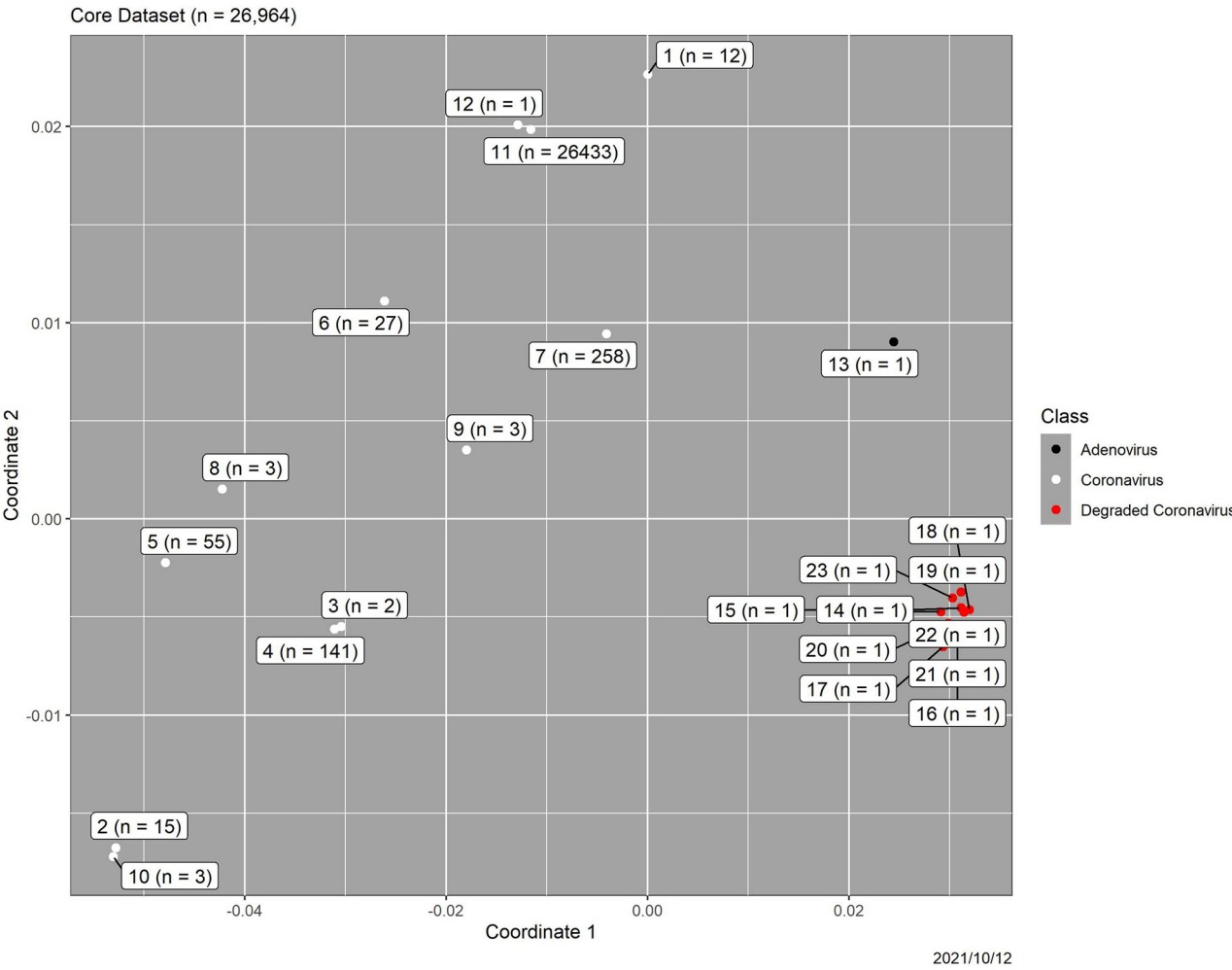

**Fig 6. Results for triplet distribution clustering.** Two-dimensional MDS plot of the 23 triplet distribution cluster centroids for a 26,964-element coronavirus database containing 11 deliberately added outliers. Labels are cluster numbers and counts. Greater distance implies higher dissimilarity.

coronavirus cluster in the triplet distribution analysis. As noted above, the largest cluster in the triplet entropy degradation cluster contains only 1470 genomes. The adenovirus outlier and the ten degraded coronavirus outliers are placed together in cluster 34, and Fig 8 shows that they clearly differ from all of the other genomes. Clusters 1–5 contain candidate outliers. Not only are they relatively small, but also each differs strongly from *all* of the other clusters. They are suggestively similar to clusters 1–10 for triplet distributions, which we pursue below. In Fig 8, the 11 deliberate outliers in cluster 34 are distant from the majority of the coronavirus genomes, but no more so than the 18 coronavirus genomes in cluster 2.

**Relationships between the two sets of clusters.** Clusters 1–10 in the triplet distribution analysis together contain 519 genomes, a number similar to the number of genomes in clusters 1–5 for the triplet entropy analysis. Moreover, both analyses separate the 11 deliberate outliers from the 26,953 legitimate coronavirus genomes, although differently. The triplet distribution

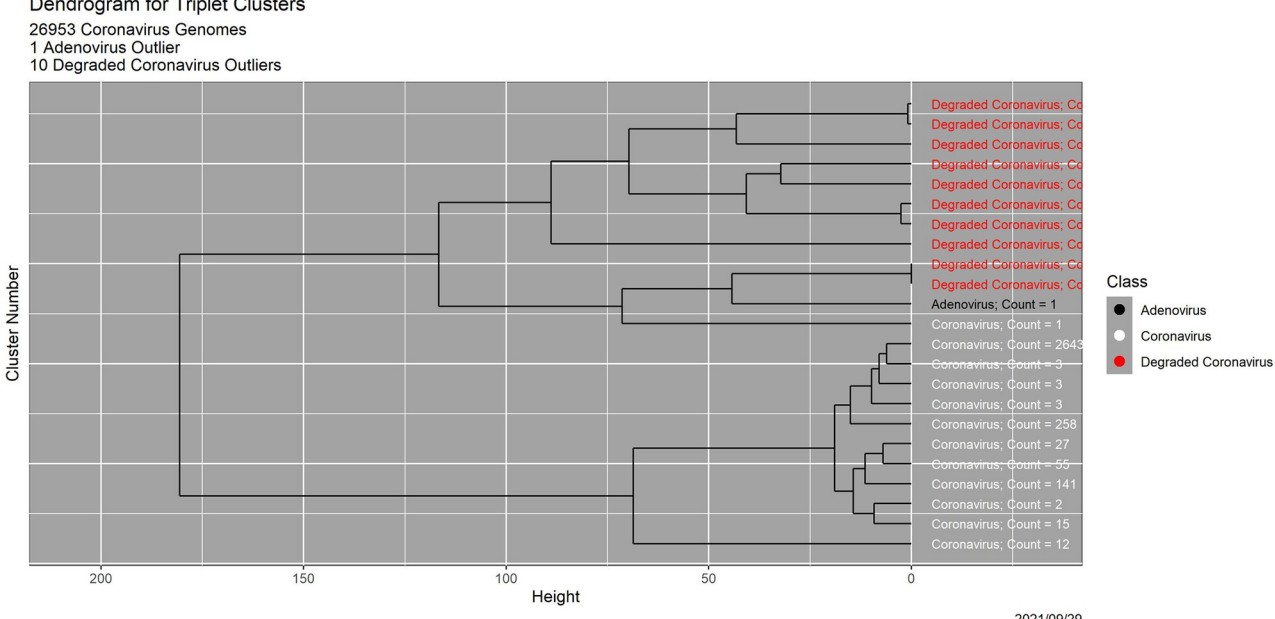

**Fig 7. Results for triplet distribution clustering.** Dendrogram, in which clusters are in order from 1 at the bottom to 23 at the top. Colors are the same as in Fig 6.

analysis places these outliers in 11 distinct clusters, while the triplet entropy degradation analysis places them all in a single cluster.

Table 1 shows the complete and strongly block-diagonal relationship between the two sets of clusters. For clarity, cells in Table 1 containing values of 0 are highlighted in pink. In detail,

1. Triplet entropy degradation cluster 34 is, as noted above, an amalgamation of triplet distribution clusters 13–24; both contain the 11 deliberate outliers.

2. The lone coronavirus genome in triplet distribution cluster 12 is absorbed into entropy degradation cluster 7, along with 843 other coronavirus genomes. Perhaps it is not an outlier after all. There is further evidence to this effect in [6]: when clustering is done using amino acid distributions, it also ceases to be an outlier. Specifically, it is merged with cluster 11 to form a massive amino acid cluster of size 26,434.

3. Triplet entropy degradation clusters 6–33 disaggregate the massive, 26,433-genome triplet distribution cluster 11, modulo four additional genomes.

4. Triplet entropy degradation clusters 1–5, containing 516 genomes and triplet distribution clusters 1–8 and 10, both containing 516 genomes, are identical collectively. These are in the upper-left corner in Table 1. Clearly, the two approaches are detecting the same outliers, with different nuances.

5. Triplet distribution cluster 9, with 3 genomes, is anomalous. Each genome it contains lies in its own large entropy degradation cluster.

Therefore, much of the scientific interpretation of outliers in [6], which is based on the text string ID variable in the NCBI database, carries over here.

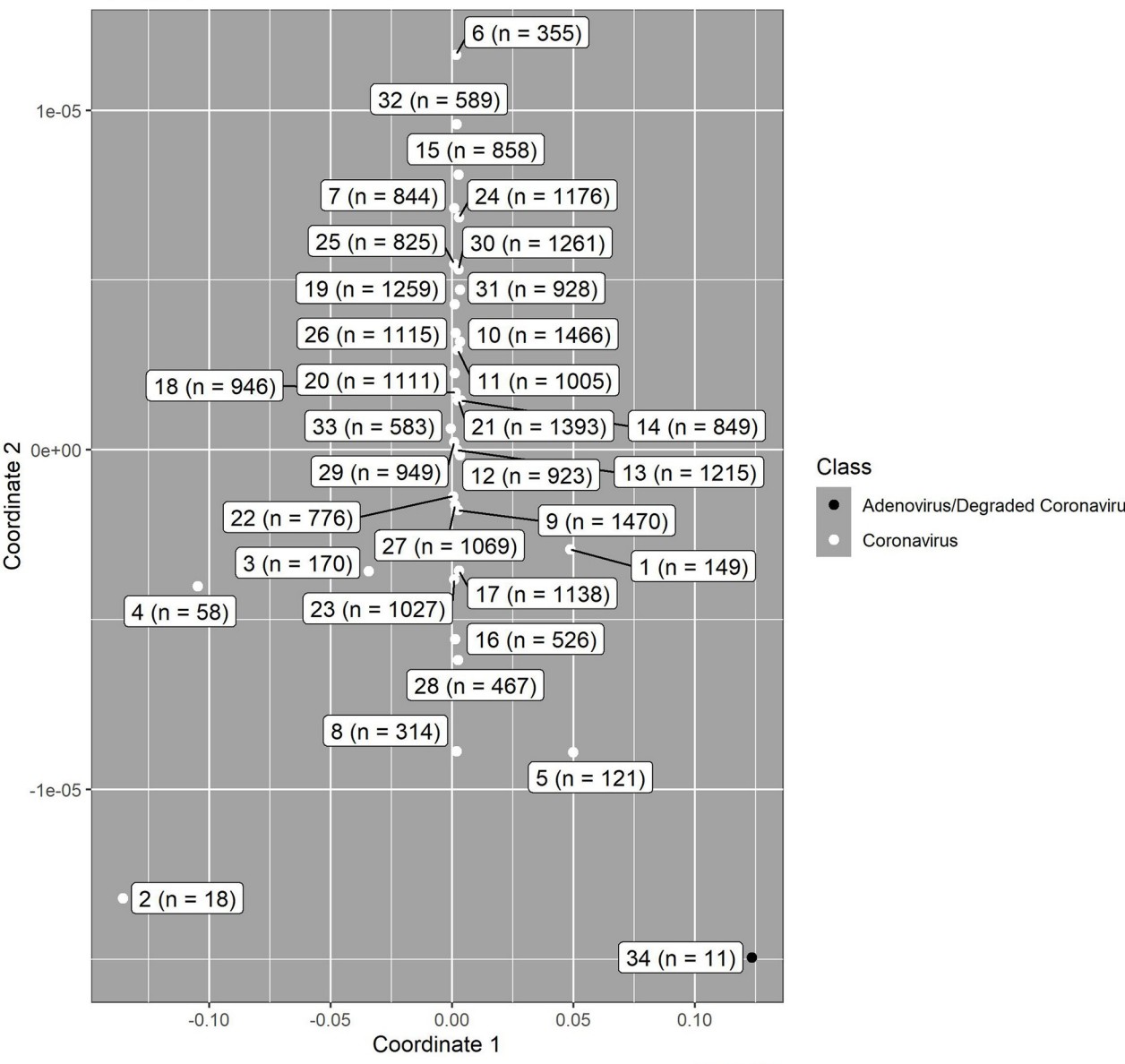

**Fig 8. Results for triplet entropy clustering.** Two-dimensional MDS plot of the 34 cluster centroids. Labels are cluster numbers and counts. Greater distance implies higher dissimilarity.

## Higher-order DNA structure

Degradation attenuates (relatively) low-dimensional genome characteristics such as tuple distributions (Fig 3). We see here that more complex structure such as repeats and palindromes is affected more strongly. As exemplar, we use an *E. coli* genome of length 4,639,675 downloaded from NCBI; the same genome appears again in Fig 10.

**Repeats.** Repeats are inherent to non-virus genomes, leading, *inter alia*, to the discovery of clustered regularly interspaced short palindromic repeats (CRISPR) in *E. coli* [17]. Table 2

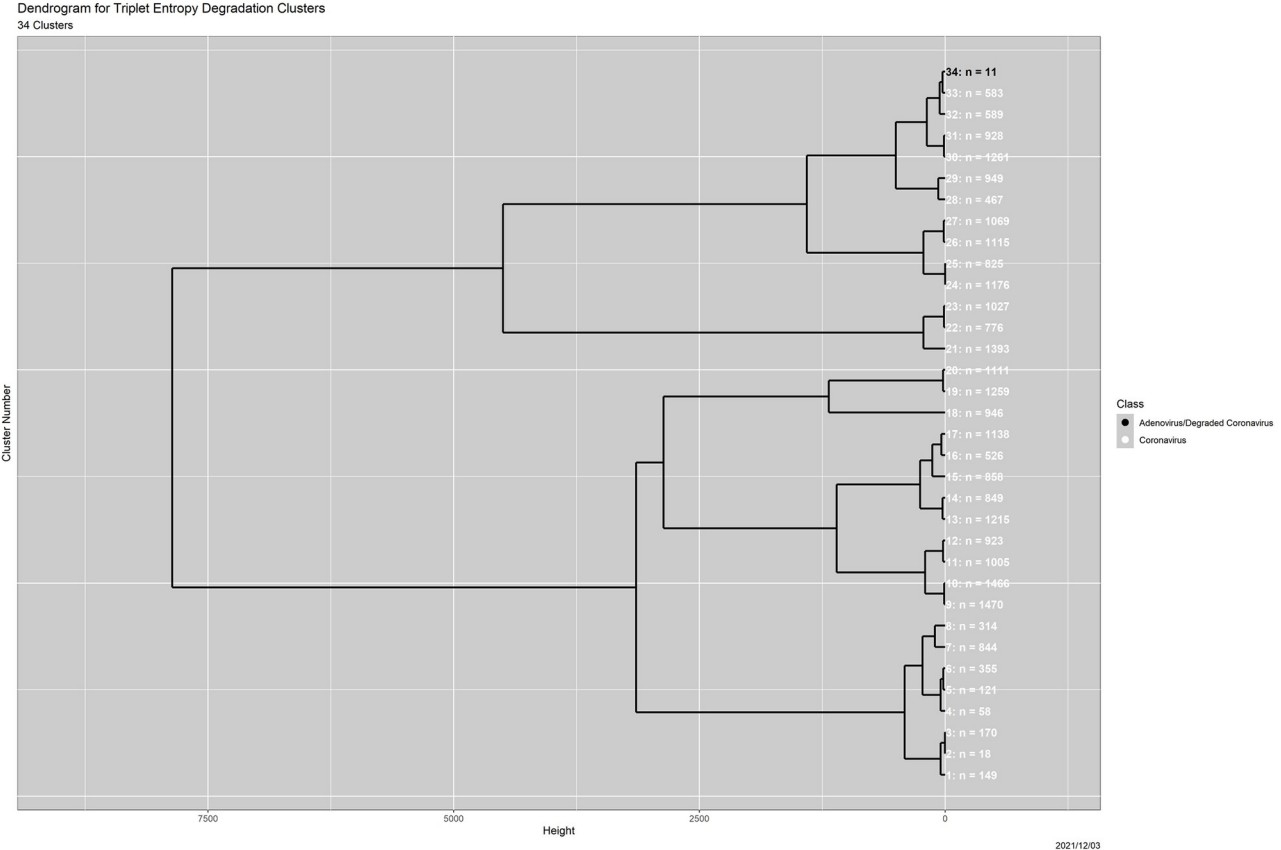

**Fig 9. Results for triplet entropy clustering.** Dendrogram, in which clusters are in order from 1 at the bottom to 34 at the top. Colors are the same as in Fig 8.

shows the effect of `Mason_variator` degradation on the numbers of repeats of various lengths in the *E. coli* genome. The column for length 29, rather than 30, honors the original discovery of CRISPR. By 500 `Mason_variator` iterations, all repeats of length 20 or longer have been obliterated. Those of length 29 are gone at 300 iterations.

**Palindromes.** Genomic palindromes, unlike those in ordinary language, consist of a sequence of bases followed immediately by its reverse complement. So, an example is ATTC-GATT‖AATCGAAT. (The ‖ has been inserted for visual clarity.) In what follows, palindromes are parameterized by half-length; the example has half-length 8. Their behavior with respect to `Mason_variator` degradation differs somewhat from that of repeats.

Table 3 shows that long palindromes (half-lengths 12, 14 and 16), are not plentiful to begin with, and, modulo noise discussed momentarily, vanish within 100 `Mason_variator` iterations. Palindromes with half-length 8 and 10 do decline in number, but do not vanish, even by 2000 iterations. Moreover, their numbers can increase, although not enormously. Palindromes of half length 6 barely diminish at all, and fluctuate substantially, The noise and the increases suggest that short palindromes differ from the other genome features discussed in this paper, and especially from repeats. They are resistant to the `Mason_variator` degradation, and can even be produced by it. This is not surprising because very short repeats are low-dimensional and may, therefore, be too short to be interesting biologically.

**Table 1. Cross-tabulation of the 34 degradation clusters (rows) and the 23 triplet distribution clusters (columns).** Cells containing zeros are shaded in red.

| Entropy Degradation Cluster | Triplet Distribution Cluster | | | | | | | | | | | | | | | | | | | | | | | |
|---|---|---|---|---|---|---|---|---|---|---|---|---|---|---|---|---|---|---|---|---|---|---|---|---|
| | 1 | 2 | 3 | 4 | 5 | 6 | 7 | 8 | 9 | 10 | 11 | 12 | 13 | 14 | 15 | 16 | 17 | 18 | 19 | 20 | 21 | 22 | 23 | Sum |
| 1 | 12 | 0 | 0 | 0 | 0 | 0 | 137 | 0 | 0 | 0 | 0 | 0 | 0 | 0 | 0 | 0 | 0 | 0 | 0 | 0 | 0 | 0 | 0 | 149 |
| 2 | 0 | 15 | 0 | 0 | 0 | 0 | 0 | 0 | 0 | 3 | 0 | 0 | 0 | 0 | 0 | 0 | 0 | 0 | 0 | 0 | 0 | 0 | 0 | 18 |
| 3 | 0 | 0 | 2 | 141 | 0 | 27 | 0 | 0 | 0 | 0 | 0 | 0 | 0 | 0 | 0 | 0 | 0 | 0 | 0 | 0 | 0 | 0 | 0 | 170 |
| 4 | 0 | 0 | 0 | 0 | 55 | 0 | 0 | 3 | 0 | 0 | 0 | 0 | 0 | 0 | 0 | 0 | 0 | 0 | 0 | 0 | 0 | 0 | 0 | 58 |
| 5 | 0 | 0 | 0 | 0 | 0 | 0 | 121 | 0 | 0 | 0 | 0 | 0 | 0 | 0 | 0 | 0 | 0 | 0 | 0 | 0 | 0 | 0 | 0 | 121 |
| 6 | 0 | 0 | 0 | 0 | 0 | 0 | 0 | 0 | 1 | 0 | 354 | 0 | 0 | 0 | 0 | 0 | 0 | 0 | 0 | 0 | 0 | 0 | 0 | 355 |
| 7 | 0 | 0 | 0 | 0 | 0 | 0 | 0 | 0 | 1 | 0 | 842 | 1 | 0 | 0 | 0 | 0 | 0 | 0 | 0 | 0 | 0 | 0 | 0 | 844 |
| 8 | 0 | 0 | 0 | 0 | 0 | 0 | 0 | 0 | 1 | 0 | 313 | 0 | 0 | 0 | 0 | 0 | 0 | 0 | 0 | 0 | 0 | 0 | 0 | 314 |
| 9 | 0 | 0 | 0 | 0 | 0 | 0 | 0 | 0 | 0 | 0 | 1470 | 0 | 0 | 0 | 0 | 0 | 0 | 0 | 0 | 0 | 0 | 0 | 0 | 1470 |
| 10 | 0 | 0 | 0 | 0 | 0 | 0 | 0 | 0 | 0 | 0 | 1466 | 0 | 0 | 0 | 0 | 0 | 0 | 0 | 0 | 0 | 0 | 0 | 0 | 1466 |
| 11 | 0 | 0 | 0 | 0 | 0 | 0 | 0 | 0 | 0 | 0 | 1005 | 0 | 0 | 0 | 0 | 0 | 0 | 0 | 0 | 0 | 0 | 0 | 0 | 1005 |
| 12 | 0 | 0 | 0 | 0 | 0 | 0 | 0 | 0 | 0 | 0 | 923 | 0 | 0 | 0 | 0 | 0 | 0 | 0 | 0 | 0 | 0 | 0 | 0 | 923 |
| 13 | 0 | 0 | 0 | 0 | 0 | 0 | 0 | 0 | 0 | 0 | 1215 | 0 | 0 | 0 | 0 | 0 | 0 | 0 | 0 | 0 | 0 | 0 | 0 | 1215 |
| 14 | 0 | 0 | 0 | 0 | 0 | 0 | 0 | 0 | 0 | 0 | 849 | 0 | 0 | 0 | 0 | 0 | 0 | 0 | 0 | 0 | 0 | 0 | 0 | 849 |
| 15 | 0 | 0 | 0 | 0 | 0 | 0 | 0 | 0 | 0 | 0 | 858 | 0 | 0 | 0 | 0 | 0 | 0 | 0 | 0 | 0 | 0 | 0 | 0 | 858 |
| 16 | 0 | 0 | 0 | 0 | 0 | 0 | 0 | 0 | 0 | 0 | 526 | 0 | 0 | 0 | 0 | 0 | 0 | 0 | 0 | 0 | 0 | 0 | 0 | 526 |
| 17 | 0 | 0 | 0 | 0 | 0 | 0 | 0 | 0 | 0 | 0 | 1138 | 0 | 0 | 0 | 0 | 0 | 0 | 0 | 0 | 0 | 0 | 0 | 0 | 1138 |
| 18 | 0 | 0 | 0 | 0 | 0 | 0 | 0 | 0 | 0 | 0 | 946 | 0 | 0 | 0 | 0 | 0 | 0 | 0 | 0 | 0 | 0 | 0 | 0 | 946 |
| 19 | 0 | 0 | 0 | 0 | 0 | 0 | 0 | 0 | 0 | 0 | 1259 | 0 | 0 | 0 | 0 | 0 | 0 | 0 | 0 | 0 | 0 | 0 | 0 | 1259 |
| 20 | 0 | 0 | 0 | 0 | 0 | 0 | 0 | 0 | 0 | 0 | 1111 | 0 | 0 | 0 | 0 | 0 | 0 | 0 | 0 | 0 | 0 | 0 | 0 | 1111 |
| 21 | 0 | 0 | 0 | 0 | 0 | 0 | 0 | 0 | 0 | 0 | 1393 | 0 | 0 | 0 | 0 | 0 | 0 | 0 | 0 | 0 | 0 | 0 | 0 | 1393 |
| 22 | 0 | 0 | 0 | 0 | 0 | 0 | 0 | 0 | 0 | 0 | 776 | 0 | 0 | 0 | 0 | 0 | 0 | 0 | 0 | 0 | 0 | 0 | 0 | 776 |
| 23 | 0 | 0 | 0 | 0 | 0 | 0 | 0 | 0 | 0 | 0 | 1027 | 0 | 0 | 0 | 0 | 0 | 0 | 0 | 0 | 0 | 0 | 0 | 0 | 1027 |
| 24 | 0 | 0 | 0 | 0 | 0 | 0 | 0 | 0 | 0 | 0 | 1176 | 0 | 0 | 0 | 0 | 0 | 0 | 0 | 0 | 0 | 0 | 0 | 0 | 1176 |
| 25 | 0 | 0 | 0 | 0 | 0 | 0 | 0 | 0 | 0 | 0 | 825 | 0 | 0 | 0 | 0 | 0 | 0 | 0 | 0 | 0 | 0 | 0 | 0 | 825 |
| 26 | 0 | 0 | 0 | 0 | 0 | 0 | 0 | 0 | 0 | 0 | 1115 | 0 | 0 | 0 | 0 | 0 | 0 | 0 | 0 | 0 | 0 | 0 | 0 | 1115 |
| 27 | 0 | 0 | 0 | 0 | 0 | 0 | 0 | 0 | 0 | 0 | 1069 | 0 | 0 | 0 | 0 | 0 | 0 | 0 | 0 | 0 | 0 | 0 | 0 | 1069 |
| 28 | 0 | 0 | 0 | 0 | 0 | 0 | 0 | 0 | 0 | 0 | 467 | 0 | 0 | 0 | 0 | 0 | 0 | 0 | 0 | 0 | 0 | 0 | 0 | 467 |
| 29 | 0 | 0 | 0 | 0 | 0 | 0 | 0 | 0 | 0 | 0 | 949 | 0 | 0 | 0 | 0 | 0 | 0 | 0 | 0 | 0 | 0 | 0 | 0 | 949 |
| 30 | 0 | 0 | 0 | 0 | 0 | 0 | 0 | 0 | 0 | 0 | 1261 | 0 | 0 | 0 | 0 | 0 | 0 | 0 | 0 | 0 | 0 | 0 | 0 | 1261 |
| 31 | 0 | 0 | 0 | 0 | 0 | 0 | 0 | 0 | 0 | 0 | 928 | 0 | 0 | 0 | 0 | 0 | 0 | 0 | 0 | 0 | 0 | 0 | 0 | 928 |
| 32 | 0 | 0 | 0 | 0 | 0 | 0 | 0 | 0 | 0 | 0 | 589 | 0 | 0 | 0 | 0 | 0 | 0 | 0 | 0 | 0 | 0 | 0 | 0 | 589 |
| 33 | 0 | 0 | 0 | 0 | 0 | 0 | 0 | 0 | 0 | 0 | 583 | 0 | 0 | 0 | 0 | 0 | 0 | 0 | 0 | 0 | 0 | 0 | 0 | 583 |
| 34 | 0 | 0 | 0 | 0 | 0 | 0 | 0 | 0 | 0 | 0 | 0 | 0 | 1 | 1 | 1 | 1 | 1 | 1 | 1 | 1 | 1 | 1 | 1 | 11 |
| Sum | 12 | 15 | 2 | 141 | 55 | 27 | 258 | 3 | 3 | 3 | 26433 | 1 | 1 | 1 | 1 | 1 | 1 | 1 | 1 | 1 | 1 | 1 | 1 | 26964 |

## Other genomes

Fig 10 demonstrates that degradation behavior is not confined to viruses. It is the analog of Fig 5 for two bacterial genomes—*P. gingivalis* and *E. coli*, for three human chromosomes—1, X and Y, and for human mitochondrial DNA. All six genomes were downloaded from NCBI; the human genome is identified as GRCh38.p13.

## Discussion

At least three paths for further research are clear. The first is that our paradigm does not yet produce quantified uncertainties about the decisions it may engender. Following this path also

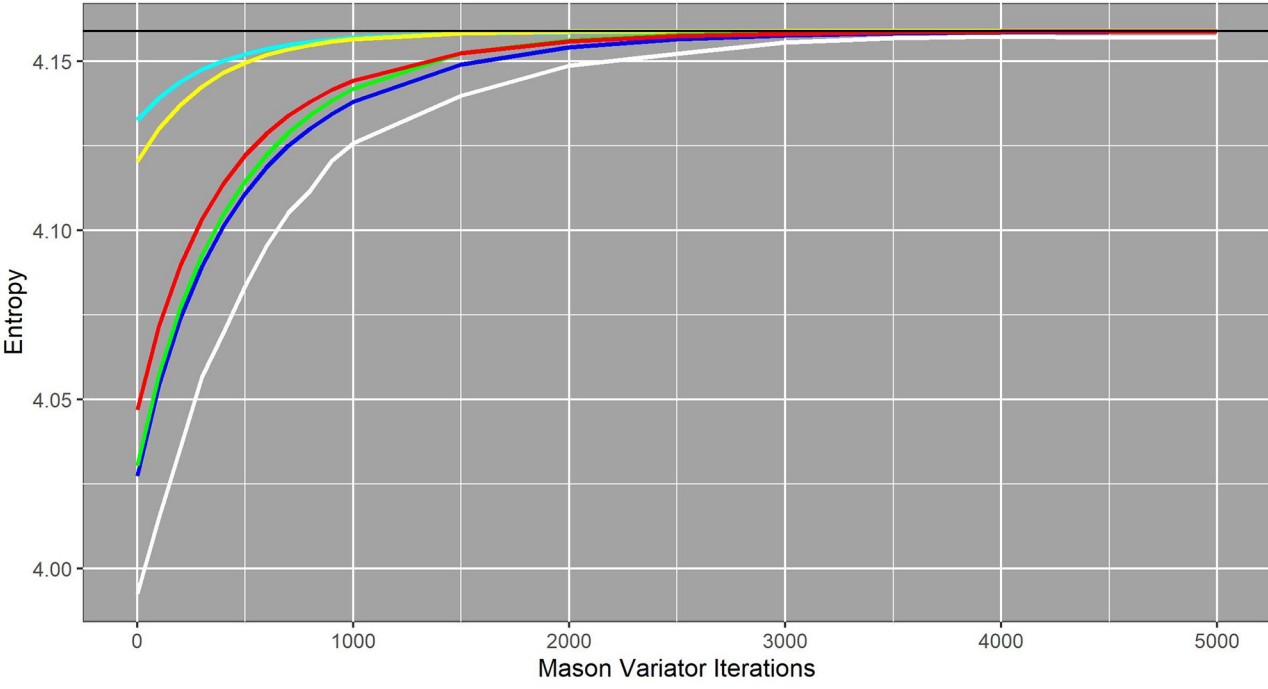

**Fig 10. Entropy as a function of `Mason_variator` iterations, for selected prokaryote and eukaryote genomes.** Included are two bacterial genomes—*P. gingivalis* and *E. coli*, three human chromosomes—1, X and Y, and human mitochondrial DNA.

**Table 2. Numbers of repeats as a function of `Mason_variator` iterations.** The genome is *E. coli*. Repeats of length 20, 25, 29, 35 and 40 are columns and `Mason_variator` iterations are rows.

| | Repeat Length | | | | |
|---|---|---|---|---|---|
| **Iteration** | **20** | **25** | **29** | **35** | **40** |
| 0 | 37285 | 35588 | 34667 | 33429 | 32530 |
| 100 | 2865 | 1094 | 487 | 153 | 58 |
| 200 | 640 | 156 | 47 | 11 | 2 |
| 300 | 152 | 11 | 0 | 0 | 0 |
| 400 | 36 | 1 | 0 | 0 | 0 |
| 500 | 17 | 0 | 0 | 0 | 0 |
| 600 | 14 | 0 | 0 | 0 | 0 |
| 700 | 6 | 0 | 0 | 0 | 0 |
| 800 | 8 | 0 | 0 | 0 | 0 |
| 900 | 10 | 0 | 0 | 0 | 0 |
| 1000 | 6 | 0 | 0 | 0 | 0 |

**Table 3. Numbers of palindromes as a function of `Mason_variator` iterations.** The genome is *E. coli*. Half lengths of 6, 8, 10, 12, 14 and 16 are columns and `Mason_variator` iterations are rows.

| | Half-Length | | | | | |
|---:|---:|---:|---:|---:|---:|---:|
| **Iterations** | **6** | **8** | **10** | **12** | **14** | **16** |
| 0 | 1128 | 113 | 22 | 11 | 2 | 1 |
| 100 | 1149 | 102 | 7 | 1 | 0 | 0 |
| 200 | 1163 | 90 | 4 | 1 | 0 | 0 |
| 300 | 1209 | 87 | 6 | 1 | 0 | 0 |
| 400 | 1137 | 81 | 6 | 1 | 0 | 0 |
| 500 | 1114 | 79 | 5 | 0 | 0 | 0 |
| 600 | 1141 | 79 | 8 | 0 | 0 | 0 |
| 700 | 1126 | 81 | 8 | 0 | 0 | 0 |
| 800 | 1130 | 75 | 4 | 1 | 1 | 0 |
| 900 | 1140 | 67 | 3 | 1 | 1 | 0 |
| 1000 | 1077 | 62 | 4 | 0 | 0 | 0 |
| 1500 | 1104 | 66 | 2 | 0 | 0 | 0 |
| 2000 | 1072 | 68 | 3 | 0 | 0 | 0 |

requires, as raised in S1 Appendix, more explicit attention to the decisions to be made using the data. To consider decision quality fully leads to the second path—better understanding of the effects of data quality on bioinformatics software pipelines. Now that we can create data of demonstrably and quantifiably lower quality, this path is feasible. Third and more speculatively, there is the relationship between data quality and adversarial attacks on genome databases or software pipelines [4, 18, 19]. Attempts to "pollute" databases with (what may turn out to be) low quality genomes are potentially detectable using the outlier identification strategies presented here. Risk-utility paradigms for statistical disclosure limitation (see S1 Appendix) are relevant, especially the need to distinguish attackers from legitimate users of the data.

Finally, there is clear potential to extend our paradigm to contexts other than genomics, provided that a credible generative model for quality degradation can be constructed. To illustrate, in the official statistics context, one simply needs a mechanism to simulate one of more forms of total survey error.

## Conclusions

In this paper, we have introduced and investigated a new, degradation-based approach to data quality for genome sequence databases, and established that it is sound scientifically and statistically. Our principal application is to outlier detection, and our methods are demonstrably effective.

## Supporting information

**S1 Appendix. Background on data quality.**
(PDF)

## Author Contributions

**Conceptualization:** Alan F. Karr, Jason Hauzel, Adam A. Porter, Marcel Schaefer.

**Formal analysis:** Alan F. Karr.

**Investigation:** Alan F. Karr, Marcel Schaefer.

**Methodology:** Alan F. Karr, Jason Hauzel.

**Project administration:** Adam A. Porter, Marcel Schaefer.

**Resources:** Adam A. Porter.

**Software:** Jason Hauzel, Adam A. Porter.

**Validation:** Alan F. Karr.

**Visualization:** Alan F. Karr.

**Writing – original draft:** Alan F. Karr.

**Writing – review & editing:** Alan F. Karr.

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
