## [Decision Letter · Decision Letter 0]

14 Apr 2022

PONE-D-22-04115Measuring Quality of DNA Sequence Data via DegradationPLOS ONE

Dear Dr. Karr,

Thank you for submitting your manuscript to PLOS ONE. After careful consideration, we feel that it has merit but does not fully meet PLOS ONE’s publication criteria as it currently stands. Therefore, we invite you to submit a revised version of the manuscript that addresses the points raised during the review process.

We look forward to receiving your revised manuscript.

Kind regards,

Alvaro Galli

Academic Editor

PLOS ONE

Journal Requirements:

2. In the Methods section of your manuscript, please ensure you provide all information needed for a reader to locate the same data sources used in this study. This includes information specifically identifying the genomes used in this study, and information on how to locate the coronavirus database used in this study.

"This research was supported in part by NIH grant 5R01AI100947–06, “Algorithms and Software for the Assembly of Metagenomic Data,” to the University of Maryland College Park (Mihai Pop, PI)"

We note that you have provided funding information. However, funding information should not appear in the Acknowledgments section or other areas of your manuscript. We will only publish funding information present in the Funding Statement section of the online submission form. 

"All authors were supported by National Institutes of Health grant 5R01AI100947--06, "Algorithms and Software for the Assembly of Metagenomic Data," to the University of Maryland College Park (Mihai Pop, PI) , via a subaward to Fraunhofer USA. The sponsor URL is www.nih.gov. The sponsor played no role in the research, decision to publish, or preparation of the manuscript."

Reviewers' comments:

Reviewer's Responses to Questions

**Comments to the Author**

1. Is the manuscript technically sound, and do the data support the conclusions?

Reviewer #1: Yes

2. Has the statistical analysis been performed appropriately and rigorously? 

Reviewer #1: I Don't Know

3. Have the authors made all data underlying the findings in their manuscript fully available?

Reviewer #1: Yes

4. Is the manuscript presented in an intelligible fashion and written in standard English?

Reviewer #1: Yes

5. Review Comments to the Author

Reviewer #1: Dear author

This is an interesting topic. However there are some points I would like to address:

1- in page 3, you have referred the figures as follows

File = Figure2Top.tif

File = Figure2Middle.tif

File = Figure2Middle.tif

Figure2middle is duplicated, and the figure 2 bottom is not mentioned.

2- is there copyright/ permission to use the figures from the Maison variator? please indicate.

3- please include line numbers

4- for the equation, use MathType for display and inline equations, as it will provide the most reliable outcome. If this is not possible, Equation Editor or Microsoft's Insert→Equation function is acceptable.

5- Footnotes are not permitted. If your manuscript contains footnotes, move the information into the main text or the reference list, depending on the content.

6- write full affiliation including ORCID when applicable , remove them from footnotes.

7- the title needs to be more clear.

8- I have seen a lot of self referencing:

Karr, A. F. (2013). Discussion of five papers on “Systems and Architectures for High-Quality Statistics

Production”. Journal of Official Statistics, 29(1):157–163.

Karr, A. F. and Cox, L. H. (2012). The World’s Simplest Survey Microsimulator (WSSM). Technical

Report 181, National Institute of Statistical Sciences, Research Triangle Park, NC. Available online at

http://www.niss.org/sites/default/files/tr181.pdf.

Karr, A. F., Hauzel, J., Menon, P., Porter, A. A., and Schaefer, M. (2021a). Specified Certainty Classification.

Technical report, Fraunhofer Center Mid-Atlantic, Riverdale, MD. arXiv/2109.06677.

13

Karr, A. F., Hauzel, J., Porter, A. A., and Schaefer, M. (2021b). Application of Markov Structure of

Genomes to Outlier Identification and Read Classification. Technical report, Fraunhofer Center MidAtlantic, Riverdale, MD.

Karr, A. F., Kohnen, C. N., Oganian, A., Reiter, J. P., and Sanil, A. P. (2006a). A framework for evaluating

the utility of data altered to protect confidentiality. The American Statistician, 60(3):224–232.

Karr, A. F., Sanil, A. P., and Banks, D. L. (2006b). Data quality: A statistical perspective. Statistical

Methodology, 3(2):137–173.

Karr, A. F., Sanil, A. P., Sacks, J., and Elmagarmid, E. (2001). Workshop Report: Affiliates Workshop on Data Quality. Technical Report, National Institute of Statistical

please include in the introduction this if this is a continuous research.

9- the data quality section should be moved after the reference heading.

10 - use vancuver style of referencing and in cite referencing. the references should be in Arabic numbers (1,2,3,4,5,6...) and not in alphabetical order.

11- short and long title cannot be the same.

6. PLOS authors have the option to publish the peer review history of their article (what does this mean?). If published, this will include your full peer review and any attached files.

Reviewer #1: **Yes: **Noora R. Al-Snan, PhD

---

## [Author Response · Author response to Decision Letter 0]

21 Jun 2022

We thank the reviewers and editors for their comments. Specific responses are as follows:

Formatting: The entire manuscript has been converted to the PLOSOne LATEX template, with the

correct section titles and the bibliography entries included in the source file

DataQualityViaDataDegradation_PLOSOne_Revised.tex. Footnotes have

been eliminated. The previous appendix has been converted supplementary material, namely

S1 Appendix. Because of the scope of the changes and use of LATEX, no “tracked changes”

file can sensibly be created.

Figures: Because figure numbering has changed, a completely new set of TIFF files has been

uploaded.

Self-referencing: has been reduced dramatically.

Funding Statement: Funding information has been removed from the manuscript. No changes are

necessary to the Funding Statement.

Data availability: Data files are now publicly available at

https://dataverse.harvard.edu/dataset.xhtml

?persistentId=doi%3A10.7910%2FDVN%2FQ6HVFO&version=DRAFT.

ORCID ID I believe that I have entered it successfully. However, in case there are any issues, it is

0000-0002-7253-0129, and I give my permission to add it. ORCID IDs for all authors who

have them appear in the manuscript

---

## [Decision Letter · Decision Letter 1]

12 Jul 2022

Measuring Quality of DNA Sequence Data via Degradation

PONE-D-22-04115R1

Dear Dr. Karr,

We’re pleased to inform you that your manuscript has been judged scientifically suitable for publication and will be formally accepted for publication once it meets all outstanding technical requirements.

Kind regards,

Alvaro Galli

Academic Editor

PLOS ONE

Additional Editor Comments (optional):

Reviewers' comments:

Reviewer's Responses to Questions

**Comments to the Author**

1. If the authors have adequately addressed your comments raised in a previous round of review and you feel that this manuscript is now acceptable for publication, you may indicate that here to bypass the “Comments to the Author” section, enter your conflict of interest statement in the “Confidential to Editor” section, and submit your "Accept" recommendation.

Reviewer #1: All comments have been addressed

2. Is the manuscript technically sound, and do the data support the conclusions?

Reviewer #1: Yes

3. Has the statistical analysis been performed appropriately and rigorously? 

Reviewer #1: I Don't Know

4. Have the authors made all data underlying the findings in their manuscript fully available?

Reviewer #1: Yes

5. Is the manuscript presented in an intelligible fashion and written in standard English?

Reviewer #1: Yes

6. Review Comments to the Author

Reviewer #1: Dear author ,

Thank you for addressing all the comments. However, I have one comment only

1- please clearly state who is the corresponding author along with the email address.

once again, thank you for addressing all the comments

regards

7. PLOS authors have the option to publish the peer review history of their article (what does this mean?). If published, this will include your full peer review and any attached files.

Reviewer #1: **Yes: **Noora Al-Snan

---

## [Editor Report · Acceptance letter]

25 Jul 2022

PONE-D-22-04115R1 

Measuring quality of DNA sequence data via degradation 

Dear Dr. Karr:

I'm pleased to inform you that your manuscript has been deemed suitable for publication in PLOS ONE. Congratulations! Your manuscript is now with our production department. 

Kind regards, 

on behalf of

Dr. Alvaro Galli 

Academic Editor

PLOS ONE